# Patterns of compensatory mutations in *rpoA*/*B*/*C* genes of multidrug resistant *M. tuberculosis* in Uganda

David Patrick Kateete[1,2*], Shakira Namakula[1,2�], Edgar Kigozi[1,2�], Fred A. Katabazi[1,2], George William Kasule[3], Kenneth Musisi[3], Edward Wampande[1,4], Deus Lukoye[3], Moses L. Joloba[1,2*]

**1** Department of Medical Microbiology, School of Biomedical Sciences, Makerere University College of Health Sciences, Kampala, Uganda, **2** Department of Immunology and Molecular Biology, School of Biomedical Sciences, Makerere University College of Health Sciences, Kampala, Uganda, **3** National TB/Leprosy Program, Ministry of Health Uganda, Kampala, Uganda, **4** Department of Veterinary Medicine, Clinical and Comparative Medicine, College of Veterinary Medicine, Animal Resources and Bio-Security, Makerere University, Kampala, Uganda

☉ These authors contributed equally to this work
* david.kateete@mak.ac.ug (DPK), mlj10@case.edu (MLJ)

## Abstract

Mutations in *rpoB*, a gene that encodes the bacterial RNA polymerase (RNAP) beta-subunit, can cause high-level resistance to rifampicin. Approximately 95% of rifampicin-resistant *Mycobacterium tuberculosis* clinical isolates harbour mutations in an 81-base pair *rpoB* region referred to as the rifampicin-resistance-determining region (*rpoB*/RRDR). Also, rifampicin-resistant *M. tuberculosis* clinical isolates carry multiple mutations in RNAP genes (i.e., *rpoA, rpoB, rpoC, rpoD*), particularly *rpoA* and *rpoC,* which encode the alpha- ($\alpha_2$) and beta'- (β') subunits, respectively. Such secondary mutations offset the fitness cost associated with acquisition of rifampicin-resistance mutations in *M. tuberculosis*, resulting in resistant strains that are as fit as the wild-type drug-susceptible strains. To analyse the patterns of compensatory mutations in RNAP encoding genes of rifampicin-resistant *M. tuberculosis* clinical isolates in Uganda, whole genome sequencing and Sanger DNA sequencing were performed on 52 *M. tuberculosis* clinical isolates – 20 drug-susceptible and 32 multidrug resistant (MDR). A total of 24 (75%) MDR-TB isolates had high-level rifampicin-resistance-conferring mutations in *rpoB*/RRDR, i.e., Ser531Leu (31%); His526Asp (6%); His526Leu (3%); His526Tyr (3%); His526Arg (3%); His526Gly (3%); Asp516Tyr (13%); Asp516Val (6%); Glu513Lys (3%); Leu511Pro (3%); Leu492Leu (3%); Gln490Arg (3%). Further, two putative compensatory mutations (Gln490Arg & Lys1025Glu) outside the RRDR and not resistance-conferring were found in *rpoB*. Altogether, 15 (63%, 15/24) MDR-TB isolates with *rpoB*/RRDR resistance-conferring mutations had non-synonymous mutations in *rpoC* of the following patterns Leu39Phe (3%); Tyr61His (3%); Asp271Gly (3%); Ser377Ala (3%); Pro481Thr (3%);

**Data availability statement:** The data generated and analysed in this study are included in this published article (and its Supplementary S files, S1 and S2 Tables). The raw sequence data was deposited in the NCBI BioProject, and the SRA records are accessible with accession number PRJNA1337100 or by following the link: https://www.ncbi.nlm.nih.gov/sra/PRJNA1337100.

**Funding:** The author(s) received no specific funding for this work.

**Competing interests:** The authors have declared that no competing interests exist.

Val483Ala (6%); Leu516Pro (3%); Ala521Asp (3%); Gly594Glu (13%); Asn698Ser (3%); Leu823Pro (3%). In conclusion, putative compensatory mutations are prevalent in rifampicin-resistant *M. tuberculosis* clinical isolates in Uganda, with *rpoC*/Gly-594Glu and *rpoC*/Val483Ala as the most frequent. Further studies will determine their association with strain genetic background, fitness and transmission in an endemic setting with a high burden of HIV-TB coinfection.

## Introduction

Multidrug resistance (MDR) in tuberculosis (TB), caused by *Mycobacterium tuberculosis* bacteria that are resistant to the two powerful anti-TB drugs (i.e., rifampicin and isoniazid [1,2]), is a major threat to global TB control efforts and the attainment of the United Nations' Sustainable Development Goal (SDG) #3 in the low- and middle-income countries (LMICs). Uganda is a TB endemic LMIC with a high burden of HIV and TB coinfection, and relatively low MDR-TB rates, i.e., ~1.4% and ~12.1% prevalence in newly diagnosed cases and retreated patients, respectively [3].

Bacterial RNA polymerase (RNAP) is an essential enzyme comprising five subunits, i.e., $\alpha_2\beta\beta'\omega$ that are responsible for DNA-dependent RNA synthesis [1,4,5]. The $\alpha_2\beta\beta'\omega$ pentameric core forms a crab-claw-like structure where the alpha subunits ($\alpha_2$) are responsible for assembly while the $\beta\beta'$ heterodimer forms the catalytic centre [6]. Unlike eukaryotic genomes that encode three distinct RNA polymerases, prokaryotes use one polymerase to synthesize rRNA, mRNA and tRNA, making the bacterial RNAP a fitness-determining enzyme as growth is dependent on the rate of production of rRNA [4,6].

Rifampicin inhibits bacterial RNAP by binding to the beta ($\beta$) subunit, disrupting the DNA/RNA channel and blocking elongation of the RNA transcripts, which is bactericidal [1,5,6]. As such, mutations in *rpoB*, a gene that encodes the bacterial RNAP $\beta$-subunit, can cause high-level resistance to rifampicin; approximately 95% of *M. tuberculosis* clinical isolates resistant to rifampicin possess a non-synonymous mutation in an 81 base pair (bp) region of *rpoB* commonly referred to as the rifampicin-resistance-determining region (RRDR; *rpoB*/RRDR), and such mutations often confer high-level resistance to rifampicin [1,2,7,8]. Whilst antibiotic resistance in bacteria is associated with lower Darwinian fitness in the absence of antibiotics, clinical isolates of rifampicin-resistant bacteria do not suffer a fitness deficit and often have levels of fitness similar to, or higher than, those of wild-type drug-susceptible bacteria [1,9]. (Here, 'fitness' refers to 'the ability of a pathogenic bacterium to establish an infection, replicate and persist in an infected host, and its capacity to be efficiently transmitted' to more susceptible hosts [10]). Furthermore, clinical isolates of rifampicin-resistant *M. tuberculosis* carry multiple mutations in the RNAP encoding genes (i.e., *rpoA*, *rpoB*, *rpoC*, *rpoD* [*rpoA/B/C/D*]) [1,5], particularly *rpoA* and *rpoC*, which encode the alpha ($\alpha_2$) and $\beta'$ subunits, respectively) [1,5,6]. It has been experimentally and/or epidemiologically demonstrated that these secondary mutations in *RNAP* genes offset the fitness cost associated with acquisition of rifampicin-resistance mutations

in *M. tuberculosis*, resulting in drug-resistant bacteria that are as fit as the wildtype drug-susceptible strains [1,5,6,8,9]. So far, it has been proven that compensatory mutations in *M. tuberculosis* (i) exhibit convergent evolution and commonly occur in rifampicin-resistant clinical isolates with resistance-conferring mutations in the *rpoB*/RRDR, (ii) do not occur in rifampicin-susceptible clinical isolates nor in rifampicin-resistant clinical isolates without resistance-conferring mutations in the *rpoB*/RRDR [1,6,8–14] and, (iii) tend to be more prevalent in regions with the highest MDR-TB incidence in the world (e.g., countries in Southern Africa and Eastern Asia and/or Europe, e.g., South Africa, China, Russia, Abkhazia/Georgia, Uzbekistan, Kazakhstan, etc.) [1,6,9,12–15].

As yet, little is known about the patterns of RNAP compensatory mutations among rifampicin-resistant *M. tuberculosis* clinical isolates from Uganda, a TB-endemic country with a high burden of both TB and HIV/AIDS. Here, we studied the *rpoA/B/C/D* gene sequences from clinical isolates of MDR- and pan-susceptible *M. tuberculosis* from patients in Uganda, for patterns of mutations associated with rifampicin-resistance. Characterization of such mutations contributes to our understanding of the microbiologic factors underlying antimicrobial resistance emergence, particularly the compensatory mechanisms allowing drug-resistant *M. tuberculosis* to overcome antibiotic pressure and become highly transmissible.

## Materials and methods

### Study setting and isolates

This retrospective cross-sectional study was conducted on stored TB cultures at Makerere University College of Health Sciences in Kampala, Uganda, between 24th June, 2019, and 17th April, 2020. A total of 52 *M. tuberculosis* clinical isolates were studied, of which 20 were drug-susceptible and 32 were MDR-TB isolates, i.e., rifampicin-resistant and isoniazid-resistant. The clinical and demographic characteristics of the patients, sputum sampling and TB culturing procedures were described previously in the national and Kampala drug-resistance surveys [3,16]; briefly, the isolates were collected from a nationally representative cohort of new TB cases and previously treated sputum smear-positive patients registered at the TB diagnostic and treatment centres during 2009−2011 [3]. Similarly, phenotypic drug susceptibility testing for sensitivity to anti-TB drugs was described in the drug resistance surveys [3,16] however, repeat culturing and susceptibility testing using the Löwenstein-Jensen (L-J) proportional method was done to validate the susceptibility patterns. *M. tuberculosis* culturing and drug sensitivity testing were performed in a BSL-3 Mycobacteriology Laboratory at the Department of Medical Microbiology, Makerere University School of Health Sciences.

### Extraction of chromosomal DNA

Stored bacterial isolates were accessed for sub-culturing on 1st July, 2019; isolates were recovered by sub-culturing on Middlebrook 7H10 agar (Becton and Dickson, USA), incubating at 37ºC in a carbon dioxide incubator (Thermal Scientific, USA), and observing daily for 28 days for growth of the bacilli. The cells were harvested and suspended in absolute ethanol (Sigma scientific, USA) to kill them by suffocation. The suspension was centrifuged at 16,000 g to obtain the cell pellet that was later re-suspended in 0.25X Tris-EDTA (TE) buffer. High-quality bacterial chromosomal DNA was extracted from *M. tuberculosis* by following the CTAB/chloroform extraction method [17]. Briefly, the cells, suspended in 0.25X TE buffer, were centrifuged at 3,000 g to wash off the media salts and residual ethanol. To lyse the cells, the cell pellet was re-suspended in 400 μL of fresh 0.25X TE buffer, followed by adding 50 μL of lysozyme (40 mg/mL) and incubating at 37°C, overnight. Then, we added 150 μL of 10% Sodium Dodecyl Sulphate (SDS)/ proteinase K solution to the mixture and incubated at 65°C for 1 hour to ascertain complete cell lysis, precipitation of proteins and other cell debris. After this, 100 μL of 5M NaCl was added to each tube, followed by 100 μL of CTAB/NaCl, and the mixture was inverted several times until the contents turned milky. To ensure complete precipitation of all cell debris the solution was incubated briefly at 65°C for 10 minutes. To purify the DNA, 750 μL of chloroform/isoamyl alcohol mixture was added to the samples and centrifuged at 16,000 g for 10 minutes. After this, the aqueous phase that contained the DNA was carefully transferred to a pre-labelled

sterile 1.5 ml microfuge tube, followed by adding 600 μL of absolute ice-cold isopropanol to precipitate the DNA. After incubating at −20°C for 2 hours, the mixture was centrifuged for 10 minutes at 16,000 g to pellet the DNA. The pellet was washed with 1 ml of 70% ice-cold ethanol and dried at room temperature for 1 hour. The DNA was eluted in 50 μL of 0.25X TE buffer. Before use in PCRs the quality and quantity of the extracted DNA were determined by electrophoresis on a 1% (w/v) agarose gel and a NanoDrop spectrophotometer (ThermoFisher Scientific).

## DNA sequencing and sequence analysis

The RNAP encoding genes (*rpoA, rpoB, rpoC, rpoD*) were PCR-amplified with previously published gene-specific primer sequences and procedures [7] for the *rpoB*/RRDR, and in-house primer sequences for *rpoA* (5'-CCGGTCACCATGTACCTACG-3' forward, 5'-GGATGTCAAGCAGGTCGGAT-3' reverse), *rpoC* (5'-CTACGTGATCACCTCGGTCG-3' forward, 5'-GTTGACGATGATTTCCGGCG-3' reverse), and *rpoD* (5'-CGATCGCGCGAAAAACCATCT-3' forward, 5'-CACCGACTGCAGT TGATCCT-3' reverse). The targeted gene segments were successfully PCR-amplified from all the study isolates. The total reaction volume for all PCRs was 60 μL, prepared according to the HotStar PCR kit (QIAGEN, Hilden, Germany). Briefly, each reaction contained 27.5 μL nuclease-free water, 6 μL 10x PCR buffer, 12 μL Q-solution, 3 μL 10mM MgCl$_2$, 3 μL of 10mM dNTPs, 1.5 μL each of reverse and forward primers, 0.5 μL High-fidelity Taq DNA polymerase (5U/μL) (Sigma-Aldrich, USA), and 10ng/μL (in 5 μL volume) of the chromosomal DNA template. Amplification was achieved in a Thermocycler (Bio-Rad Laboratories Inc., Singapore) using the following program: initial denaturation at 95°C, 5 minutes, followed by 35 cycles each consisting of 95°C, 45 seconds, 60°C, 45 seconds, and 72°C, 50 seconds, with a final extension step of 72°C, 10 minutes. Five microliters each of the PCR product was analysed using 1% agarose gel electrophoresis with Ethidium bromide 5mg/ml staining. Gels were run at 120 V for 1 hour and visualized using a UVP Gel documentation (Benchtop Trans-illuminator System_BioDoc-it, CA, USA). Then, 50 μL each of the PCR products was purified using the QIAmp DNA purification minikit (QIAGEN, Hilden, Germany), and the pure amplicons were sequenced at ACGT Inc. (Wheeling IL, USA).

The sequences obtained were analysed first through BLAST searches at NCBI https://blast.ncbi.nlm.nih.gov/Blast.cgi to confirm they significantly align to the expected gene/protein sequences in *M. tuberculosis.* To determine the amino acid substitutions, the DNA sequences were translated into amino acid sequences using the Molecular Evolutionary Genetics Analysis Version 6.0 *(*MEGA6.06) software [7] or the ExPASy online database http://web.expasy.org/translate/. To identify mutations, the sequenced amplicons were aligned in MEGA6.06 and BioEdit v7.2.5.0 to the *rpoA, rpoB, rpoC and rpoD* genes sequences from *M. tuberculosis* H37Rv (NC_000962), a reference strain pan-susceptible to anti-TB drugs. To determine whether the identified mutations were resistance-conferring, we followed the tbdream database [18] https://tbdreamdb.ki.se/Info/ and published literature [1,2,6–9,11,13,14]. All the data was curated, compiled and presented as tables or percentages depending on the frequency of mutations that occurred in the examined genes.

## Whole genome sequencing of *M. tuberculosis* genomic DNA

Since some of the clinical isolates of rifampicin-resistant *M. tuberculosis* lacked rifampicin-resistance-conferring mutations in the *rpoB*/RRDR upon analysis of the Sanger sequenced DNA amplicons, as a validation step, we performed whole genome sequencing (WGS) of all the 52 *M. tuberculosis* clinical isolates and the control strain H37Rv. WGS was performed on a MiSeq platform (Illumina) at the Genomics Unit of the Department of Immunology and Molecular Biology at Makerere University College of Health Sciences, by following the protocols published by Illumina Inc. Briefly, 20 indexed paired-end libraries were prepared starting with 0.2 ng/μL of input *M. tuberculosis* chromosomal DNA using the Nextera® XT DNA sample preparation guide (Illumina Inc., CA USA). This protocol included stages of tagmentation, indexing, normalization and library pooling before loading the pooled library into a MiSeq Reagent Kit v2 (MS-102–2003, 500 cycle) (Illumina). This kit offers an improved chemistry to increase cluster density with a maximum output of 8.5 gigabytes (GB) of data. The cartridge was loaded in the MiSeq instrument (Illumina inc.) to run a resequencing workflow for ~72 hours.

The DNA library for WGS was spiked with PhiX v3 control (Illumina) to a final concentration of 5% recommended for low diversity libraries. The generated FASTQ files were analysed and high-quality files (Phred score of ≥30) uploaded on PhyResSE, a web-based tool that delineates *M. tuberculosis* antibiotic resistance markers and their lineages using WGS data (https://bioinf.fz-borstel.de/mchips/phyresse/).

## Quality control

PCR-amplicons from *M. tuberculosis* H37Rv, a reference strain that is susceptible to all anti-TB drugs, were sequenced and included as controls. Likewise, a known MDR-TB strain that contained well-known rifampicin and isoniazid resistance-conferring mutations, earlier confirmed by the Genotype MTBDR*plus* assay (Hain Life Sciences) was included in the study as the positive control for drug resistance mutations. Negative controls for DNA extraction, PCRs, and DNA sequencing were included to eliminate contamination. High-Fidelity Taq DNA polymerase (Sigma-Aldrich, USA) which delivers superior results with a twofold higher yield and threefold greater fidelity compared to regular Taq DNA Polymerase was used in the PCRs. Sanger DNA sequencing-identified mutations were revalidated through WGS.

## Ethics statement

This study was approved by the Makerere University School of Biomedical Sciences Research and Ethics Committee (approval # SBS-HDREC-667). This committee approved the use of archived *Mycobacterium tuberculosis* isolates previously collected from patients who participated in the National Drug Resistance Survey that investigated the phenotypic levels and patterns of resistance to first- and second-line anti-TB drugs among new and previously treated sputum smear-positive TB cases in Uganda [3,16]. The parent studies stated herein had previously obtained written informed consent from the participants for sample storage and use of stored samples in further studies. The data were analysed anonymously and the authors did not have access to information that could identify individual participants during or after data collection.

## Results and discussion

We characterized a total of 52 *M. tuberculosis* clinical isolates; 32 MDR and 20 pan-susceptible to anti-TB drugs, for patterns of mutations in the *rpoA*/*B*/*C*/*D* genes associated with rifampicin-resistance in bacteria. The isolates were collected during the two drug resistance surveys conducted in the period between 2009 and 2011 [3,16], and they are representative of the patients registered at the TB treatment centres at the time [3]. Overall, using both WGS and Sanger DNA sequencing, we did not find mutations or single-nucleotide polymorphisms (SNPs) in *rpoA* and *rpoD* of all the isolates; likewise, there were no mutations/SNPs in any of the genes (*rpoA*/*B*/*C*/*D*) of the 20 drug-susceptible *M. tuberculosis* isolates and the control strain (H37Rv). As outlined below, mutations were found in the *rpoB* and *rpoC* genes in some of the MDR-TB isolates, and these were verified through WGS and Sanger DNA sequencing. In addition to rifampicin, high-level resistance-conferring mutations to other anti-TB drugs were found, i.e., in *katG* (isoniazid-resistance), *pncA* (pyrazinamide-resistance), and *rrs* (kanamycin-resistance) but are not the focus of this study; these were reported previously [15] and are summarized in S1 Table.

### *rpoB* mutations

Fig 1 and Table 1 summarize the patterns of mutations and SNPs we found in *rpoB*. A total of 24 MDR-TB isolates (75%, 24/32) had mutations in the *rpoB*/RRDR, i.e., Lys1025Glu (3%, 1/32); Ser531Leu (31%, 10/32); His526Asp (6%, 2/32); His526Leu (3%, 1/32); His526Tyr (3%, 1/32); His526Arg (3%, 1/32); His526Gly (3%, 1/32); Asp516Tyr (13%, 4/32); Asp516Val (6%, 2/32); Glu513Lys (3%, 1/32); Leu511Pro (3%, 1/32); Leu492Leu (3%, 1/32); and Gln490Arg (3%, 1/32), and these are similar to mutations that have been reported before [2,7,10,15] (note that three isolates had double

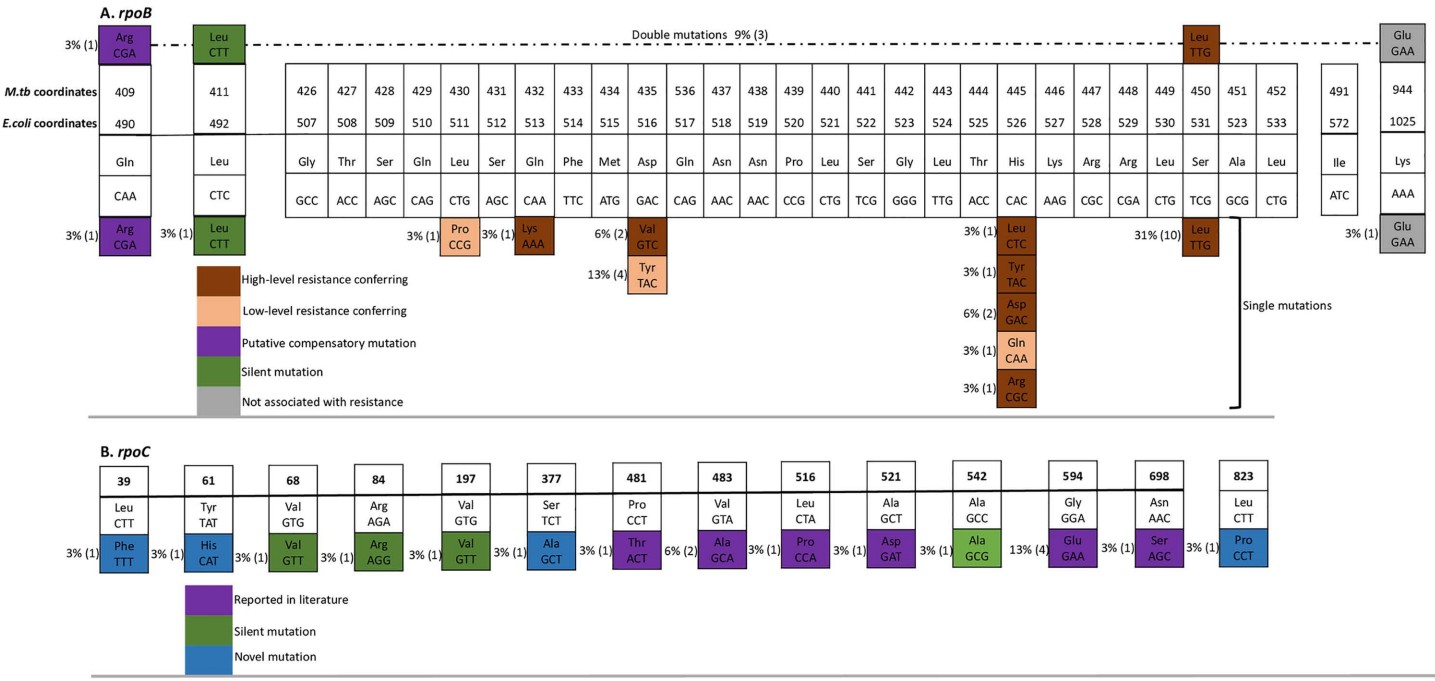

**Fig 1. Rifampicin-resistance-conferring mutations in *rpoB* (panel A) and putative compensatory mutations in *rpoB* (panel A) and *rpoC* (panel B) of MDR *M. tuberculosis* clinical isolates.**

mutations – see ahead). With the exception of Lys1025Glu, Leu492Leu and Gln490Arg, all mutations occurred in the *rpoB*/RRDR and confer high-level resistance to rifampicin [2,8] apart from Leu511Pro, Asp516Tyr, and His526Gln that are categorized as low-level rifampicin-resistance-conferring mutations [2,8]. Therefore, apart from two mutations (i.e., Glu513Lys and Leu511P), rifampicin-resistance in this study is attributed to mutations at three notable codons – 516, 526, and 531, which is in line with reports from many settings around the world [2,7]; notably, there is high polymorphism at codon 526 compared to the other two codons (516 and 531), **Fig 1**.

Furthermore, three isolates (#s 3, 19, 20) were double mutants (Table 1, Fig 1) in that each harboured two mutations i.e., Asp516Val & Lys1025Glu; Leu492Leu & Ser531Leu; and Gln490Arg & Ser 531Leu, respectively. Asp516Val and Ser-531Leu are high-level rifampicin-resistance-conferring mutations (see above) while the other three mutations in the double mutants occurred outside of the *rpoB*/RRDR and are not likely to be resistance-conferring [2]; non-synonymous *rpoB* polymorphisms like Gln490Arg and Lys1025Glu that are outside of the *rpoB*/RRDR and are not resistance-conferring have been reported to have compensatory effects [6] hence, are putative compensatory mutations.

### *rpoC* mutations and co-occurrence with mutations in *rpoB*/RRDR

A total of 16 MDR-TB isolates (50%, 16/32) had mutations in *rpoC* – fifteen non-synonymous i.e., Leu39Phe (3%, 1/32); Tyr61His (3%, 1/32); Asp271Gly (3%, 1/32); Ser377Ala (3%, 1/32); Pro481Thr (3%, 1/32); Val483Ala (6%, 2/32); Leu-516Pro (3%, 1/32); Ala521Asp (3%, 1/32); Gly594Glu (13%, 4/32); Asn698Ser (3%, 1/32); Leu823Pro (3%, 1/32) and four synonymous, i.e., Val68Val (3%, 1/32); Arg84Arg (3%, 1/32); Val197Val (3%, 1/32); and Ala542Ala (3%, 1/32). Hence, the most frequent *rpoC* mutations were Gly594Glu (13%) and Val483Ala (6%), **Fig 1** & **Table 1**. Of note, mutations Pro481Thr, Val483Ala, Leu516Pro, Ala521Asp, Gly594Glu and Asn698Ser are the most frequently reported *rpoC* compensatory muta-tions in rifampicin-resistant *M. tuberculosis* clinical isolates in many settings around the world [1,6,8–14].

**Table 1. Mutations in the *rpoB* and rpo*C* genes of the MDR-TB clinical isolates.**

| No. | Sample/ isolate ID | *M. tuberculosis* lineage/ strain/ genotype | *rpoB* (*Rv0667*) RRDR | | *rpoC* (Rv0668) | |
| --- | --- | --- | --- | --- | --- | --- |
| | | | Codon change | Mutation (*M. tuberculosis* coordinates) | Codon change | Mutation |
| 1. | A_ND9010 | 4/Uganda | TCG→TTG | Ser531Leu (S450L) | CTA→CCA | Leu516Pro |
| 2. | B_ND5532 | 2/Beijing | TCG→TTG | Ser531Leu (S450L) | CTA→CCA | Leu516Pro |
| 3. | C_ND8803B | 4/Uganda | GAC→GTC | Asp516Val (D435V) | CTT→TTT | Leu39Phe |
| | | | | | TAT→CAT | Tyr61His |
| | | | | | GTG→GTT | Val68Val |
| | | | | | TCT→GCT | Ser377Ala |
| | | | AAA→GAA | Lys1025Glu (K944E) | | |
| 4. | D_ND473 | 4/EAM | GAC→TAC | Asp516Tyr (D435Y) | GGA→GAA | Gly594Glu |
| 5. | E_ND1001 | 4/EAM | GAC→TAC | Asp516Tyr (D435Y) | GGA→GAA | Gly594Glu |
| 6. | F_ND6332 | 4/LAM | TCG→TTG | Ser531Leu (S450L) | GTA→GCA | Val483Ala |
| 7. | G_ND6493 | 2/Beijing | TCG→TTG | Ser531Leu (S450L) | GCT→GAT | Ala521Asp |
| 8. | H_ND6907 | 4/Uganda | CAA→AAA | Gln513Lys (Q432K) | CCT→ACT | Pro481Thr |
| 9. | I_ND7389 | 2/Beijing | TCG→TTG | Ser531Leu (S450L) | AAC→AGC | Asn698Ser |
| 10. | J_ND6166 | 3/Delhi/CAS | CAC→CTC | His526Leu (H445L) | – | |
| 11. | K_ND6332 | 4/LAM | TCG→TTG | Ser531Leu (S450L) | GTA→GCA | Val483Ala |
| 12. | L_ND5677 | 4/EAM | CAC→TAC | His526Tyr (H445Y) | – | |
| 13. | M_ND1000 | 4/EAM | CAC→GAC | His526Asp (H445D) | – | |
| 14. | N_ND6927 | 4/EAM | GAC→TAC | Asp516Tyr (D435Y) | GGA→GAA | Gly594Glu |
| 15. | O_ND5XXX | 4/Uganda | TCG→TTG | Ser531Leu (S450L) | AGA→AGG | Arg84Arg |
| 16. | P_ND5171 | 3/Delhi/CAS | GTG→CCG | Leu511Pro (L430P) | GAC→GGC | Asp271Gly |
| 17. | Q_ND11753 | 4/LAM | CAC→CAA | His526Gln (H445Q) | GAC→GGC | Asp271Gly |
| 18. | R_ND7415 | 4/Uganda | CAC→CGC | His526Arg (H445R) | GCC→GCG | Ala542Ala |
| 19. | S_NDXXXX | 4/Uganda | CTC→CTT | Leu492Leu (L411L) | GTT→GTG | Val197Val |
| | | | TCG→TTG | Ser531Leu (S450L) | | |
| 20. | T_ND7776 | 4/Uganda | CAA→CGA | Gln490Arg (Q409R) | CTT→CCT | Leu823Pro |
| | | | TCG→TTG | Ser531Leu (S450L) | | |
| 21. | U_ND7508 | 4/LAM | TCG→TTG | Ser531Leu (S450L) | – | |
| 22. | V_ND8803 | 4/Uganda | GAC→GTC | Asp516Val (D435V) | – | |
| 23. | W_ND6927 | 4/EAM | CAC→GAC | His526Asp (H445D) | – | |
| 24. | X_ND5730 | 4/EAM | GAT→TAT | Asp516Tyr (D435Y) | GGA→GAA | Gly594Glu |
| 25. | Y1_ND70865 | 4/Uganda | – | | – | |
| 26. | Y2_ND70865 | 2/Beijing | – | | – | |
| 27. | Y3_ND70865 | 4/LAM | – | | – | |
| 28. | Y4_ND70865 | 3/Delhi/CAS | – | | – | |
| 29. | Y5_ND70865 | 3/Delhi/CAS | – | | – | |
| 30. | Y6_ND70865 | 2/Beijing | – | | – | |
| 31. | Y7_ND70865 | 4/EAM | – | | – | |
| 32. | Y8_ND70865 | 4/S | – | | – | |
| | | 4/H37Rv (control) | – | | – | |

- No mutation detected; EAM, Euro-American super-lineage; LAM, Latin American Mediterranean; CAS, Central Asian Strain.

While most of the mutations have been previously reported, we identified four new polymorphisms i.e., Leu39Phe, Tyr61His, Asp271Gly, and Ser377Ala, **Fig 1** & **Table 1**, which are likely to be compensatory mutations since the RNAP encoding genes are highly conserved in bacteria particularly *M. tuberculosis* [1,9,19]. Furthermore, 15 (63%) of the 24 MDR-TB isolates with rifampicin-resistance conferring mutations in *rpoB*/RRDR had non-synonymous mutations in *rpoC*; in line with current evidence that compensatory mutations do occur only in rifampicin-resistant clinical isolates with rifampicin-resistance conferring mutations in *rpoB*/RRDR [1,9,12–14], we did not find mutations/SNPs in *rpoC* of the eight MDR-TB isolates that lacked rifampicin-resistance-conferring mutations, **Table 1**.

Overall, in this study, the proportion of MDR-TB isolates with putative compensatory mutations in *rpoC* i.e., 66.7% (16/24) is high compared to rates from other countries [1,8,9,12–14], and the fact that MDR-TB prevalence is low in Uganda [20,21]; however, other investigators reported comparable or higher rates, e.g., Wang et al in China [13] and Vargas et al in Peru [12] reported that 98.2% (54/55) and 54% (95/175) respectively, of rifampicin-resistant *M. tuberculosis* isolates with resistance-conferring mutations in *rpoB*/RRDR had putative compensatory nonsynonymous mutations in *rpoA/rpoC*.

Furthermore, almost all isolates (90%, 9/10) with the *rpoB*/Ser531Leu resistance-conferring mutation harboured putative compensatory mutations in *rpoC* (**Table 1**), and similar observations for this mutation have been reported [8–11,13,14]. Furthermore, Gly594Glu, the most prevalent *rpoC* mutation in our study (**Table 1**) and reported before as a putative compensatory mutation [1,6,8–14], occurred among both rifampicin-resistant and rifampicin-susceptible isolates in Peru [12]. This discrepancy could be attributed to issues related to phenotypic drug susceptibility testing approaches i.e., the L-J proportional method used in the current study [3,16] vs. slide drug susceptibility testing (SDST) in the Peruvian study [12]. Generally, phenotypic susceptibility testing methods, especially SDST, could yield false-negative results [22] for isolates with low-level resistance-conferring mutations leading to mis-identification of resistant isolates as susceptible. Furthermore, the synonymous mutation Ala542Ala was previously reported as a lineage-defining polymorphism (i.e., phylogenetic marker) for the Latin American Mediterranean (LAM) sub-lineage of the Euro-American *M. tuberculosis* super-family [9]; however, in this study, we did not detect Ala542Ala in LAM strains but instead was detected in *M. tuberculosis* sub-lineage Uganda, **Table 1**. This discrepancy could be related to the sensitivity of genotyping approaches used i.e., Spoligotyping/*IS6110* RFLP fingerprinting in the South African study [9], which is inferior and has a low discriminatory power compared to DNA sequencing used in the current study [23].

One limitation in our study is that we had fewer rifampicin-resistant TB isolates, since Uganda as a country has low MDR-TB rates. However, our findings have implications for the understanding of fitness, transmissibility and antimicrobial resistance mechanisms in a TB endemic setting with high HIV-TB coinfection rates. Further studies in Uganda could consider compensatory mutations as potential factors contributing to strain fitness or increased transmission of resistant TB strains. This is particularly important in the context of strain genetic background, given that Uganda has a predominant strain causing majority of TB disease in the country, i.e., *M. tuberculosis* sub-lineage Uganda, a Lineage 4 (EAM) sub-clade [24–26]. In addition to Lineage 4, *M. tuberculosis* comprises seven other major genetic clades, i.e., Lineages 1, 2, 3, 5, 6, 7 and 8, which are distributed differently around the world, and this has led to local adaptation of the pathogen [27] and infecting one quarter of the global human population (approximately 2 billion people) despite evidence of the host mounting a strong immune response to TB infection [28,29]. Little is known about the mechanisms underlying the predominance of the *M. tuberculosis* sub-lineage Uganda strains in Uganda. Compensatory mutations could play a role, occurring more selectively in rifampicin-resistant sub-lineage Uganda strains compared to other genotypes, conferring more fitness hence, improved host adaptability and/or transmissibility.

## Conclusions

This study shows that putative compensatory mutations are prevalent in clinical isolates of rifampicin-resistant *M. tuberculosis* in Uganda with *rpoC*/Gly594Glu and *rpoC*/Val483Ala as the most frequent; also, for the first time, the study identifies putative compensatory mutations in *rpoB* (Gln490Arg and Lys1025Glu) of rifampicin-resistant *M. tuberculosis* in Uganda. Further

studies are required to investigate the association of such mutations with the strain genetic background, as well as their effect on strain fitness and TB transmission in an endemic setting with high HIV-TB coinfection rates but low prevalence of MDR-TB.

## Supporting information

**S1 Table. Resistance mutations – this table provides a catologue of high-level resistance conferring mutations to other anti-TB drugs we detected i.e., in the *katG* gene (isoniazid resistance), *pncA* gene (pyrazinamide resistance) and *rrs* gene (kanamycin resistance).**
(XLSX)

**S2 File. Bam (Binary Alignment/Map) files – these are compressed, binary versions of the Sequence Alignment/Map (SAM) files used to store nucleotide sequence alignments against the H37Rv *M. tuberculosis* reference genome.**
(ZIP)

## Acknowledgments

We thank the contribution and support given by staff at the Departments of Medical Microbiology and Immunology & Molecular Biology at the School of Biomedical Sciences, Makerere University College of Health Sciences in Kampala, Uganda.

## Author contributions

**Conceptualization:** David Patrick Kateete.

**Data curation:** Edgar Kigozi, Edward Wampande.

**Formal analysis:** David Patrick Kateete, Shakira Namakula, Edgar Kigozi, Fred A. Katabazi, George William Kasule, Deus Lukoye.

**Funding acquisition:** David Patrick Kateete, Moses L. Joloba.

**Investigation:** Edgar Kigozi, Fred A. Katabazi, George William Kasule, Deus Lukoye.

**Methodology:** David Patrick Kateete, Shakira Namakula, Edgar Kigozi, Deus Lukoye.

**Project administration:** Edgar Kigozi, Moses L. Joloba.

**Resources:** Kenneth Musisi.

**Supervision:** David Patrick Kateete.

**Visualization:** Edward Wampande.

**Writing – original draft:** David Patrick Kateete.

**Writing – review & editing:** David Patrick Kateete, Shakira Namakula, Edgar Kigozi, George William Kasule, Kenneth Musisi, Deus Lukoye, Moses L. Joloba.

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
