## [Decision Letter · Decision Letter 0]

17 Aug 2025

Dear Dr. Kateete,

Thank you for submitting your manuscript to PLOS ONE. After careful consideration, we feel that it has merit but does not fully meet PLOS ONE’s publication criteria as it currently stands. Therefore, we invite you to submit a revised version of the manuscript that addresses the points raised during the review process.

We look forward to receiving your revised manuscript.

Kind regards,

Muhammad Qasim, Ph.D

Academic Editor

PLOS ONE

Journal Requirements:

2. In the online submission form, you indicated that the data underlying the results presented in the study are available from the corresponding authors (DPK & MLJ).

Research reported in this publication was supported by the Fogarty International Center of the National Institutes of Health under Award Number D43TW010319; the project was also supported in part by the EDCTP2 programme of the European Union (grant number TMA2018CDF-2357-MTI-Plus) to DPK. The content is solely the responsibility of the authors and does not necessarily represent the official views of the funders.

Reviewers' comments:

Reviewer's Responses to Questions

**Comments to the Author**

1. Is the manuscript technically sound, and do the data support the conclusions?

Reviewer #1: Yes

Reviewer #2: Yes

2. Has the statistical analysis been performed appropriately and rigorously?

Reviewer #1: Yes

Reviewer #2: Yes

3. Have the authors made all data underlying the findings in their manuscript fully available?

Reviewer #1: Yes

Reviewer #2: Yes

4. Is the manuscript presented in an intelligible fashion and written in standard English?

Reviewer #1: Yes

Reviewer #2: Yes

Reviewer #1: The manuscript is very well written and the data very valuable. There are minors remarks to address and improve the manuscript.

Here below the remarks:

1- In lines 271-273: “Furthermore, 16 (67%) of the 24 MDR-TB isolates with rifampicin-resistance conferring mutations in rpoB/RRDR had non-synonymous mutations in rpoC (one had a synonymous mutation)”, it is not clear if 16 or 15 of the 24 isolates with mutations in RRDR had non-synonymous rpoC mutations? In the Table 1 the isolates that have non-synonymous rpoC mutations are the 1, 2, 3, 4, 5, 6, 7, 8, 9, 11, 14, 16, 17, 20 and 24 that make in total 15 isolates. The isolates 15, 18 and 19 have synonymous rpoC mutations and the isolates 10, 12, 13, 21, 22 and 23 don’t have rpoC mutations. It is recommended to revise the numbers and confirm if this is correct.

2- In the Table 1, the division of the lines 16 and 17 for the columns rpoC is missing.

3- In lines 278 to 284 it is said that 50% (16/32) is the proportion of MDR-TB isolates with rpoC mutations and compared to other studies such as Wang et al in China and Vargas et al in Peru that reported 98.2% (54/55) and 54% (95/175) respectively. However, the last authors calculated the proportion using as denominator the rifampicin resistant M. tuberculosis isolates with resistance-conferring mutations in rpoB/RRDR. To compare the proportions the authors of this manuscript should use as denominator the 24 isolates with resistance-conferring mutations in rpoB/RRDR. The proportion would be 16/24 (66.7%).

4- In the introduction the authors said that the characterization of the compensatory mutations contributes to the understanding of the microbiologic factors underlying antimicrobial resistance emergence, particularly the compensatory mechanisms allowing dug-resistant MTB to overcome drug pressure and become highly transmissible. To what extent the characterization of the rpoC mutations in Uganda contributed to the understanding of the emergence of resistance and transmission of the MDR isolates? Explore more about the added value of the data analysed and conclusions.

Reviewer #2: The manuscript can be accepted after making minor corrections.

Authors should analyze the correlation of recent mutation trends in high-burden countries worldwide, as this analysis is currently lacking.

**Do you want your identity to be public for this peer review?** For information about this choice, including consent withdrawal, please see our Privacy Policy

Reviewer #1: No

Reviewer #2: **Yes: ** Muthuraj Muthaiah

---

## [Author Response · Author response to Decision Letter 1]

7 Oct 2025

JOURNAL REQUIREMENTS

Response:

We have revised and formatted the manuscript in accordance with the guidance in the above links. To the best of our knowledge we ensured that the revised version of the manuscript meets PLOS ONE's style requirements, including those for file naming.

2. In the online submission form, you indicated that the data underlying the results presented in the study are available from the corresponding authors (DPK & MLJ). All PLOS journals now require all data underlying the findings described in their manuscript to be freely available to other researchers, either 1. In a public repository, 2. Within the manuscript itself, or 3. Uploaded as supplementary information. This policy applies to all data except where public deposition would breach compliance with the protocol approved by your research ethics board. If your data cannot be made publicly available for ethical or legal reasons (e.g., public availability would compromise patient privacy), please explain your reasons on resubmission and your exemption request will be escalated for approval.

Response:

We thank you for pointing this out. We followed options 1 and 3 as guided and provided the data as Suplementary files (S1 and S2 Tables) and also deposited the raw sequencing data (SRAs) in in a public repository (https://www.ncbi.nlm.nih.gov/sra/PRJNA1337100 - NCBI), which is now accessible to the public. We have updated the manuscript by adding a ‘data availability subsection’ in the revised version, see lines 209-213.

3. Thank you for stating the following in the Acknowledgments Section of your manuscript: Research reported in this publication was supported by the Fogarty International Center of the National Institutes of Health under Award Number D43TW010319; the project was also supported in part by the EDCTP2 programme of the European Union (grant number TMA2018CDF-2357-MTI-Plus) to DPK. The content is solely the responsibility of the authors and does not necessarily represent the official views of the funders.

Please remove any funding-related text from the manuscript and let us know how you would like to update your Funding Statement. Currently, your Funding Statement reads as follows: The author(s) received no specific funding for this work.

Response:

We thank you for pointing out this anomaly. We have removed the funding information from the acknowldgement section (see lines 343-346). Though, the funding that supported this work only supported data collection and analysis, and we still lack funds to meet the publication fees including open access charges. In light of this, we kindly request that in the event that the Funding Statement is updated, it should not suppose or imply that we have funds to pay for publication fees. As we are based at an institution in Uganda, a Low- and middle-income country (LMIC), we request that the automatic waiver that was initially granted to us be maintained. We thank you.

Response:

None of the reviewer comments indicated a recommendation to cite specific previously published works.

Response:

We have reviewed the reference list and confirm that it is complete and correct. We have not cited papers that have been retracted.

REVIEWERS' COMMENTS

REVIEWER #1:

We thank Reviewer #1 for finding our manuscript to be very well written and the data valuable, and for pointing out errors and/or issues, which we have addressed in the revised manuscript as you guided; a point-by-point response is also provided below;

Comment:

1- In lines 271-273: “Furthermore, 16 (67%) of the 24 MDR-TB isolates with rifampicin-resistance conferring mutations in rpoB/RRDR had non-synonymous mutations in rpoC (one had a synonymous mutation)”, it is not clear if 16 or 15 of the 24 isolates with mutations in RRDR had non-synonymous rpoC mutations? In the Table 1 the isolates that have non-synonymous rpoC mutations are the 1, 2, 3, 4, 5, 6, 7, 8, 9, 11, 14, 16, 17, 20 and 24 that make in total 15 isolates. The isolates 15, 18 and 19 have synonymous rpoC mutations and the isolates 10, 12, 13, 21, 22 and 23 don’t have rpoC mutations. It is recommended to revise the numbers and confirm if this is correct.

Response:

Thank you for noting this error – indeed it was 15 of the 24 isolates with resistance conferring mutations in the rpoB/RRDR that had non-synonymous mutations in rpoC. We have revised the text accordingly, see lines 283-285.

Comment:

2- In the Table 1, the division of the lines 16 and 17 for the columns rpoC is missing.

Response:

The division of the lines 16 and 17 for the columns rpoC is now indicated in Table 1.

Comment:

3- In lines 278 to 284 it is said that 50% (16/32) is the proportion of MDR-TB isolates with rpoC mutations and compared to other studies such as Wang et al in China and Vargas et al in Peru that reported 98.2% (54/55) and 54% (95/175) respectively. However, the last authors calculated the proportion using as denominator the rifampicin resistant M. tuberculosis isolates with resistance-conferring mutations in rpoB/RRDR. To compare the proportions the authors of this manuscript should use as denominator the 24 isolates with resistance-conferring mutations in rpoB/RRDR. The proportion would be 16/24 (66.7%).

Response:

We agree. The proportion is 16/24 (66.7%) and we have updated the text accordingly, see lines 289-295.

Comment:

4- In the introduction the authors said that the characterisation of the compensatory mutations contributes to the understanding of the microbiologic factors underlying antimicrobial resistance emergence, particularly the compensatory mechanisms allowing dug-resistant MTB to overcome drug pressure and become highly transmissible. To what extent the characterisation of the rpoC mutations in Uganda contributed to the understanding of the emergence of resistance and transmission of the MDR isolates? Explore more about the added value of the data analysed and conclusions.

Response:

We have expounded on the added value of the data analysed, see Discussion section, lines 318-334. Briefly, our study of compensatory mutations lays a foundation for future studies in Uganda aiming to understand the pathogen factors underlying the transmission of multi-drug resistant TB. Such studies should consider compensatory mutations as potential microbiologic / pathogen factors contributing to bacterial fitness; this is particularly important given that we have a dominant TB strain in Uganda (‘M. tuberculosis Uganda’ genotype, a sub-clade in the Euro-American lineage 4 superfamily) causing majority of the TB disease in the country. One possibility / hypothesis is that compensatory mutations occur more easily in rifampicin-resistant M. tuberculosis strains with this genetic background compared to others, confer increased fitness to the bacteria, hence more transmission. We thank you.

REVIEWER #2:

The manuscript can be accepted after making minor corrections. Authors should analyze the correlation of recent mutation trends in high-burden countries worldwide, as this analysis is currently lacking.

Response:

We thank Reviewer 2 for finding our manuscript publishable in PLOS ONE. Indeed, we have described and reported our findings in the view of the latest mutation trends in the high-burden countries across the world, with comparison of findings with specific TB endemic countries in Africa, Asia and Eastern Europe. Thank you.

Response:

We have uploaded our figure file to the Preflight Analysis and Conversion Engine (PACE) digital diagnostic tool, and submitted a file version of the figure that matches the guidelines for formatting figures. Thank you.

---

## [Editor Report · Decision Letter 1]

25 Nov 2025

Patterns of compensatory mutations in rpoA/B/C genes of multidrug resistant M. tuberculosis in Uganda

PONE-D-25-36891R1

Dear Dr. Kateete,

We’re pleased to inform you that your manuscript has been judged scientifically suitable for publication and will be formally accepted for publication once it meets all outstanding technical requirements.

Kind regards,

Muhammad Qasim, Ph.D

Academic Editor

PLOS ONE
---

## [Editor Report · Acceptance letter]

PONE-D-25-36891R1

PLOS ONE

Dear Dr. Kateete,

I'm pleased to inform you that your manuscript has been deemed suitable for publication in PLOS ONE. Congratulations! Your manuscript is now being handed over to our production team.

Kind regards,

on behalf of

Dr. Muhammad Qasim

Academic Editor

PLOS ONE